# *Myristica fragrans* Extract Inhibits Platelet Desialylation and Activation to Ameliorate Sepsis-Associated Thrombocytopenia in a Murine CLP-Induced Sepsis Model

**DOI:** 10.3390/ijms24108863

**Published:** 2023-05-16

**Authors:** Seong-Hun Jeong, Ji-Young Park, Young Bae Ryu, Woo Sik Kim, In-Chul Lee, Ju-Hong Kim, Dohoon Kim, Ji-Hye Ha, Ba-Wool Lee, Jiyoung Nam, Kyoung-Oh Cho, Hyung-Jun Kwon

**Affiliations:** 1Functional Biomaterial Research Center, Korea Research Institute of Bioscience and Biotechnology, Jeongeup 56212, Republic of Korea; 2Laboratory of Veterinary Pathology, College of Veterinary Medicine, Chonnam National University, Gwangju 61186, Republic of Korea; 3Center for Companion Animal New Drug Development, Jeonbuk Branch, Korea Institute of Toxicology, Jeongeup 53212, Republic of Korea

**Keywords:** sepsis, sepsis-associated thrombocytopenia, sialidase, *Myristica fragrans*, CLP, platelet desialylation, platelet activation, Ashwell–Morell receptor

## Abstract

Sepsis, characterized by an uncontrolled host inflammatory response to infections, remains a leading cause of death in critically ill patients worldwide. Sepsis-associated thrombocytopenia (SAT), a common disease in patients with sepsis, is an indicator of disease severity. Therefore, alleviating SAT is an important aspect of sepsis treatment; however, platelet transfusion is the only available treatment strategy for SAT. The pathogenesis of SAT involves increased platelet desialylation and activation. In this study, we investigated the effects of *Myristica fragrans* ethanol extract (MF) on sepsis and SAT. Desialylation and activation of platelets treated with sialidase and adenosine diphosphate (platelet agonist) were assessed using flow cytometry. The extract inhibited platelet desialylation and activation via inhibiting bacterial sialidase activity in washed platelets. Moreover, MF improved survival and reduced organ damage and inflammation in a mouse model of cecal ligation and puncture (CLP)-induced sepsis. It also prevented platelet desialylation and activation via inhibiting circulating sialidase activity, while maintaining platelet count. Inhibition of platelet desialylation reduces hepatic Ashwell–Morell receptor-mediated platelet clearance, thereby reducing hepatic JAK2/STAT3 phosphorylation and thrombopoietin mRNA expression. This study lays a foundation for the development of plant-derived therapeutics for sepsis and SAT and provides insights into sialidase-inhibition-based sepsis treatment strategies.

## 1. Introduction

Sepsis is a potentially fatal condition in which organ dysfunction occurs due to dysregulation of the host immune response to infections [1]. Despite extensive research efforts spanning over two decades, sepsis is still the leading cause of death in critically ill patients and remains a major human health problem worldwide [2]. Thrombocytopenia, a condition characterized by low platelet count, is frequently observed in patients with sepsis. This condition increases the risk of bleeding and disseminated intravascular coagulation (DIC) and elevates levels of cytokines [3,4]. In sepsis, thrombocytopenia is an indicator of disease severity, and it increases sepsis-related morbidity and mortality [5,6,7,8]. Therefore, new treatment options are required for sepsis-associated thrombocytopenia (SAT).

Platelets are small (1–3 μm) cell fragments that lack a nucleus and are produced in the cytoplasm of megakaryocytes [9]. Circulating platelet levels should be tightly regulated because low platelet levels are associated with spontaneous bleeding, whereas high levels are linked to an increased risk of arterial blockage and potential organ damage. In a recent study, marked thrombocytopenia in *Streptococcus pneumoniae* sepsis was caused by the removal of desialylated platelets via neuraminidase A (NanA) through the Ashwell–Morell receptor (AMR) [10,11]. Sialic acid coating of the platelet membrane prevents platelet destruction. Sialidase, also known as neuraminidase, is found abundantly in viruses, bacteria, and mammalian cells; it cleaves sialic acid in platelet glycoproteins [12]. The AMR in hepatocytes recognizes and removes desialylated platelets and increases thrombopoietin (TPO) synthesis via the activation of JAK2 and STAT3 [10,13]. Platelet clearance due to desialylation can be avoided through using sialidase inhibitors such as oseltamivir phosphate (OS) [14,15].

Platelets are activated during sepsis, leading to microvascular thrombosis and organ failure [16]. The activation and entrapment of platelets within the pulmonary capillaries contribute to inflammation and coagulation in the lungs [17,18]. Additionally, activated platelets combine with neutrophils to promote the formation of extracellular traps (NETs). Prolonged and excessive NET formation leads to intravascular thrombosis and organ damage [19,20]. The activation of platelets is intensified by secondary mediators released by platelets, such as adenosine diphosphate (ADP) and thromboxane A_2_ (TXA_2_) [5]. Sialidase can cause platelet hyper-reactivity, which amplifies ADP pathway-dependent platelet activation [21]. Thus, the inhibition of microbial sialidase activity in sepsis may be a potential strategy to prevent platelet activation and thrombocytopenia.

The seed of *Myristica fragrans* (MF), nutmeg, is widely used as a spice and medicine [22]. It exerts an array of biologically beneficial effects such as anti-inflammatory, antibacterial, antioxidant, and anti-obesity effects [23,24,25], as well as analgesic, cardioprotective, hepatoprotective, and other pharmacological effects [26,27,28,29]. However, the effects of MF on thrombocytopenia during sepsis remains unexplored. We previously reported the efficacy of MF as a natural sialidase inhibitor [30]. We hypothesized that MF controls SAT through blocking sialidase-mediated platelet desialylation. In this study, we aimed to examine the effects of MF on the desialylation and activation of washed platelets. We further investigated the protective role of MF against SAT in a mouse model of cecal ligation and puncture (CLP)-induced sepsis, focusing on the JAK2/STAT3 signaling pathway in the liver.

## 2. Results

### 2.1. Chemical Profiling of M. fragrans

*Myristica fragrans* was subjected to phytochemical analysis using high-performance liquid chromatography (HPLC), and 12 compounds (1–12) in the ethanolic extract were tentatively identified (Appendix A): 6 phenylpropenes (1–6), 4 neolignans (7–10), and 2 diarylnonanoids (11 and 12) (Table 1).

### 2.2. Ethanolic Extract of M. fragrans Inhibits Desialylation in Platelets Exposed to Sialidase

To measure in vitro sialidase inhibition using MF and platelet desialylation level, washed platelets were exposed to sialidase and treated with either sialic acid- or galactose-specific binding lectin (Figure 1). Fluorescein-conjugated *Ricinus communis* agglutinin-I lectin (RCA-I) and *Erythrina crista-galli* lectin (ECA) were used to detect galactose on the surface of desialylated platelets using flow cytometry. The washed platelets exposed to sialidase showed increased RCA-I and ECA binding compared to unexposed platelets, indicating their desialylation (Figure 1a,b). The binding of RCA-I and ECA increased with the concentration of sialidase but did not change at concentrations of ≥5 mU. The interaction between *Sambucus nigra* lectin (SNA) and *Maackia amurensis* lectin II (MAL-II) with platelet surface glycoproteins decreased, which suggests that sialidase removed sialic acid from the platelets (Figure 1c,d). Although the binding of SNA and MAL-II was reduced via sialidase treatment, there was no difference in their binding at different concentrations of sialidase. Based on these results, we selected 5 mU sialidase for subsequent experiments. Platelets treated with sialidase and MF exhibited reduced RCA-I binding compared with platelets treated with sialidase alone, indicating that desialylation was inhibited by MF. (Figure 1e). The sialidase inhibitor OS was used as a positive control. Ethanolic extract of *M. fragrans* and OS reduced platelet desialylation through effectively inhibiting sialidase activity.

### 2.3. Ethanolic Extract of M. fragrans Attenuates ADP-Dependent Platelet Activation through Inhibiting Sialidase-Mediated Platelet Hyper-Reactivity

Sialidase exposure results in platelet hyper-reactivity via platelet desialylation and promotes the ADP-mediated platelet activation pathway [21]. We investigated whether MF inhibits ADP-dependent platelet activation via sialidase inhibition. Washed platelets were treated with sialidase and stimulated with ADP, and the expression of P-selectin (CD62P), a marker for platelet activation, was measured using flow cytometry. Compared with untreated platelets, those treated with sialidase–ADP showed a significant increase in P-selectin expression (Figure 2a). Pretreatment with MF and OS significantly reduced the expression of p-selectin induced via sialidase–ADP in a concentration-dependent manner (Figure 2a–c). The results indicate that MF can prevent platelet activation mediated via ADP through inhibiting desialylation of the platelets.

### 2.4. Ethanolic Extract of M. fragrans Inhibits Inflammatory Cytokine Production in and Improves Survival of CLP-Induced Sepsis Mice

As MF exhibited inhibitory effects on desialylation and activation of sialidase-stimulated platelets in vitro, we tested its effects in vivo using CLP-induced septic mice. The CLP-induced sepsis model is widely accepted as the benchmark for animal sepsis models because it closely replicates human sepsis [31]. To determine whether MF and OS improve the survival of CLP mice, MF and OS were orally administered to mice and survival of CLP mice was monitored for 96 h after CLP surgery. We found that the survival rate of the sham group was 100%. However, all mice in the CLP group died within 4 days. Notably, MF (200 mg/kg) and OS (20 mg/kg) significantly improved the survival of mice by 50% and 40%, respectively (Figure 3a). Elevated levels of pro-inflammatory cytokines TNF-α and IL-6 were observed in the serum of the CLP group compared with that of the sham group. However, TNF-α and IL-6 levels significantly decreased in the MF group compared with those in the control group (Figure 3b,c). The OS group showed no significant differences in inflammatory cytokine levels compared with the CLP group. These results indicate that MF improved survival in septic mice and suppressed the inflammatory response.

### 2.5. Ethanolic Extract of M. fragrans Prevents SAT through Inhibiting Platelet Desialylation in CLP-Induced Sepsis

Next, we evaluated whether MF inhibited thrombocytopenia and desialylation of platelets in CLP-induced septic mice. The CLP group showed a significant decrease in circulating platelet counts in the blood 24 h after the CLP surgery. The MF and OS treatments inhibited the reduction in the circulating platelet counts in mice with CLP-induced sepsis (Figure 4a). The serum sialidase activity was approximately four times higher in the CLP group than in the sham group. However, the MF and OS treatments significantly reduced serum sialidase activity in the CLP mice (Figure 4b). Furthermore, we measured platelet desialylation in the CLP mice using flow cytometry. The binding of ECA and RCA-I to the platelet surface was significantly elevated in the CLP group compared to the sham group (Figure 4c). In contrast, both MF- and OS-treated mice showed significantly reduced ECA and RCA-I binding to the platelet surface (Figure 4c–e). These results indicate that platelet desialylation increased in CLP-induced sepsis and that MF prevented platelet desialylation through inhibiting sialidase activity.

### 2.6. Ethanolic Extract of M. fragrans Inhibits Platelet Activation in CLP-Induced Sepsis

To investigate the level of platelet activation during sepsis, we assessed P-selectin (CD62P) expression on the platelet surface using flow cytometry 24 h after the CLP procedure. Compared with that in the sham group, P-selectin expression on the platelet surface in the CLP group was significantly increased. In contrast, the expression of p-selectin was significantly reduced in the CLP + MF and CLP + OS groups compared with the CLP group (Figure 5a,b). The results suggest that MF exerted a suppressive effect on platelet activation in vivo as well as in vitro. Thus, these findings demonstrated that MF is a potent inhibitor of platelet activation.

### 2.7. Ethanolic Extract of M. fragrans Attenuates Organ Damage and Fibrin Deposition in CLP-Induced Sepsis

As a decrease in circulating platelet count in sepsis is associated with the microvascular thrombosis with platelet-rich thrombi, likely contributing to organ failure and ischemic complications [32,33], we investigated organ damage and fibrin deposition in mice with CLP-induced sepsis. Sepsis-induced lung and liver damage and failure are major complications that directly contribute to disease progression and mortality [34,35]. Therefore, we histologically evaluated the lung and liver tissues in all experimental mice. As expected, mice with CLP-induced sepsis showed significant necrosis, interstitial edema, inflammatory cell infiltration, and hemorrhage in the lung tissue stained with hematoxylin and eosin (H&E). However, treatment with MF and OS markedly reduced the extent of these histological changes (Figure 6a,b). In addition, CLP mice showed hepatocellular necrosis, vacuolization, and inflammatory cell infiltration in the liver. However, these pathological changes were alleviated in the liver tissues of mice in the CLP + MF and CLP + OS groups (Figure 6a,c). Next, we investigated the appearance of platelet-rich thrombi in organ microvessels, which is associated with reduced circulating platelet count, through detecting fibrin deposition using immunohistochemistry. The MF and OS treatments markedly suppressed fibrin deposition in the liver and lungs in CLP-induced septic mice (Figure 6a,d,e).

### 2.8. Ethanolic Extract of M. fragrans Regulates Hepatic JAK2/STAT3 Signaling and TPO Expression in CLP-Induced Sepsis

AMR-mediated clearance of circulating desialylated platelets upregulates hepatic JAK2-STAT3 signaling and TPO expression [13]. As platelets were desialylated in CLP-induced sepsis, we investigated whether hepatic JAK2-STAT3 signaling was affected using Western blotting. The phosphorylation of JAK2 and STAT3 significantly increased in the livers of mice in the CLP group compared with that in mice in the sham group. Interestingly, the phosphorylation of JAK2 and STAT3 in the livers of mice in the CLP + MF and CLP + OS groups was significantly reduced compared to the CLP group (Figure 7a,b). In addition, TPO mRNA expression in the liver and plasma TPO levels were investigated. The CLP group showed increased expression of hepatic TPO mRNA, whereas the MF- and OS-treated groups showed reduced hepatic TPO mRNA expression (Figure 7c). Similarly, the level of plasma TPO increased in CLP-induced sepsis, and it was suppressed through MF and OS treatments (Figure 7d). These results suggest that MF and OS modulated the AMR-mediated platelet clearance pathway through inhibiting platelet desialylation.

## 3. Discussion

The pathophysiology of sepsis is highly complicated, involving biphasic immune responses, inflammation, and the coagulation cascade [36]. In particular, platelets, which are likely to be among the first cells to respond during sepsis, are multifunctional and are closely involved in antibacterial activity, clot formation, and immune regulation. During sepsis, these platelet mechanisms might become dysregulated and maladapted, leading to organ damage [5,37]. Sepsis-associated thrombocytopenia is common in patients with sepsis and is a major prognostic marker for increased mortality [7,38]. As platelet consumption and destruction increases and their production decreases simultaneously during sepsis, SAT is likely to occur. The mechanisms associated with SAT include platelet sequestration, platelet activation, immune-mediated destruction, and DIC [16,39,40,41]. Currently, there is no known effective method to prevent thrombocytopenia, and acute platelet transfusion is the only available treatment option. Therefore, we considered the inhibition of SAT as an important treatment strategy for sepsis.

Despite numerous studies demonstrating the effectiveness of plant extracts in treating sepsis [42], to the best of our knowledge, no study has investigated the treatment of sepsis based on the sialidase-inhibitory activity of plant extracts. In this study, we demonstrated that the sialidase-inhibitory activity of MF is effective for treating sepsis and SAT. Sialidases facilitate the hydrolysis of glycoproteins and glycolipids through removing sialic acid [43]. Sialidase is a major virulence factor in several pathogens, and it has been suggested that sialidase hydrolyzes sialic acid in platelet glycoproteins during infection [11,12]. We co-incubated washed platelets with sialidase and MF in vitro and observed a significant reduction in platelet desialylation in terms of RCA-I and ECA binding. Interestingly, the sialidase-inhibitory activity of MF was similar to that of OS, an effective sialidase inhibitor that is widely used clinically [44]. These results demonstrate that treatment with MF inhibits sialidase-mediated platelet desialylation.

Platelet activation is an essential factor in the regulation of host vascular immunity and inflammation [45]. However, during sepsis, platelet activation can cause endothelial tissue damage, lead to neutrophil extracellular trap (NET) and microthrombus formation, and worsen septic coagulation and inflammation [3,46]. These interactions between inflammation and coagulation spread systemically, leading to DIC and multi-organ dysfunction syndrome [47]. Therefore, through inhibiting platelet activation, it may be possible to decrease uncontrolled coagulation and inflammatory responses in sepsis and attenuate organ damage. In the present study, platelet activation using sialidase was induced in vitro using the platelet agonist ADP. The MF and OS treatments decreased the expression of P-selectin, a marker of platelet activation, in a concentration-dependent manner, which is consistent with the finding of a previous study, that is, NanA-induced platelet desialylation induces ADP-dependent platelet hyper-reactivity [21]. Thus, MF inhibited ADP-dependent platelet activation through preventing platelet desialylation. Additionally, in a previous study, a neolignan isolated from MF exhibited antiplatelet activity through regulating the cAMP level [48]. Therefore, these results demonstrate the potential of MF as an inhibitor of platelet activation. We further validated the in vitro inhibitory effect of MF on platelet desialylation and activation in vivo using a murine CLP-induced sepsis model. The extract significantly improved the survival rate and suppressed serum levels of inflammatory cytokines in the CLP-induced sepsis model mice. Interestingly, OS improved survival but did not reduce the serum levels of inflammatory cytokines. These results indicate the anti-inflammatory activity of MF, which is consistent with the findings of a previous study [23]. Moreover, MF and OS not only effectively inhibited platelet desialylation and activation but also maintained circulating platelet counts in the CLP-induced sepsis model mice. In addition, tissue damage and fibrin deposition were reduced in the lungs and liver. These results demonstrate that treatment with MF inhibited sialidase-mediated platelet desialylation and activation as well as ameliorated thrombocytopenia and tissue damage.

A recent study reported that senile and desialylated platelets are eliminated using hepatic AMR and that hepatic AMR regulates TPO mRNA expression in the liver [13]. The expression of TPO mRNA is regulated through the phosphorylation of JAK2/STAT3 downstream of AMR [13]. This feedback mechanism has been suggested to be important for the regulation of steady-state TPO homeostasis. Furthermore, TPO supports the survival, proliferation, and differentiation of megakaryocytes, which are key regulators of platelet production [49]. Circulating TPO level tends to be elevated in sepsis and is associated with sepsis severity, because TPO promotes platelet activation [50,51,52]. In addition, studies have demonstrated that inhibition of TPO expression prevents organ damage in the CLP model of sepsis [53]. Therefore, the regulation of TPO level and JAK2 and STAT3 phosphorylation under normal and pathological conditions is important. We found that MF and OS reduced circulating TPO level and hepatic TPO expression in CLP-induced sepsis model mice. In addition, through reducing the phosphorylation of JAK2 and STAT3 in the liver, we confirmed that they are related to the hepatic AMR-mediated desialylated platelet removal mechanism. These results suggest that MF reduced hepatic AMR-mediated platelet clearance and ameliorated thrombocytopenia through inhibiting platelet desialylation during sepsis.

Our study had a few limitations. First, the 12 identified compounds of MF are expected to have various immunological functions; however, the compounds that are closely involved in the alleviation of sepsis have not been identified. Therefore, more detailed studies on the compounds responsible for these functions are needed. Second, to better understand the pharmacological characteristics of MF, additional research on the molecular mechanisms that are involved in its ability to inhibit platelet activation is needed.

## 4. Materials and Methods

### 4.1. Reagents

Tyrode-HEPES buffer was purchased from BioSolution (Suwon, Republic of Korea). Phosphate buffered saline (1×) without calcium and magnesium (#17-516F) was purchased from Lonza (Basel, Switzerland). Acid-citrate-dextrose (ACD) solution (#C3821), prostaglandin E1 (PGE1; #P5515), neuraminidase (sialidase) from *Clostridium perfringens* (#11585886001), and OS (#SML1606) were purchased from Sigma-Aldrich (St. Louis, MO, USA). *Erythrina crista-galli* lectin (#FL-1141-5), SNA (#FL-1301-2), and RCA-I (#FL-1081-5) were purchased from Vector Laboratories (Burlingame, CA, USA). *Maackia amurensis* lectin II (#21511103-1) antibody was purchased from bioWORLD (Dublin, OH, USA). Anti-CD41 (#11-0411-82) and anti-CD62P (#17-0626-82) were purchased from Thermo Fisher Scientific (Waltham, MA, USA). Anti-fibrinogen antibody (ab34269) was purchased from Abcam (Cambridge, UK). Monoclonal antibodies against JAK2 (#3230), phospho-JAK2 (#3776), STAT3 (#12640), and phospho-STAT3 (#9145) were purchased from Cell Signaling Technology (Danvers, MA, USA). Mouse Thrombopoietin Quantikine ELISA Kit (#MTP00) was purchased from R&D Systems (Minneapolis, MN, USA). Mouse TNF ELISA Set (#555268) and Mouse IL-6 ELISA Set (#555240) were purchased from BD Biosciences (San Jose, CA, USA).

### 4.2. Ethanolic Extract of M. fragrans

We previously reported the extraction of *M. fragrans* with ethanol. The constituents of *M. fragrans* ethanol extracts were analyzed using HPLC after fractionation with polar solvents [30]. Briefly, whole *M. fragrans* plants (9.6 kg) were purchased from a local market (HumanHerb Co., Ltd., Daegu, South Korea). *Myristica fragrans* was ground and extracted with 10-fold excess of ethanol for 1 week at 22 °C. The ethanol extract was dried completely (520 g) via evaporation. Column chromatography was repeated to isolate 12 compounds, which were confirmed via comparison with spectroscopic data from previous studies [30].

### 4.3. Animals

The animal care and experimental protocols were subject to review and approval by the Institutional Animal Care and Use Committee (IACUC) of Korea Research Institute of Bioscience and Biotechnology (KRIBB). The permit number for the study was KRIBB-AEC-22267. Male C57BL/6 mice aged 6–8 weeks and free from specific pathogens (SPF) were purchased from Orient Bio (Seongnam, Republic of Korea). Five mice were housed in each ventilated cage with standard bedding and were provided ad libitum access to food and water. The mice were housed in a controlled environment with a temperature of 22 °C ± 2 °C, a humidity level of 55% ± 5%, and an artificial light–dark cycle of 12 h. The mice were allowed to acclimatize to the environment for a minimum of 1 week before being used in the experiments.

The mice were randomly assigned into four groups with 10 mice per group: (1) sham, (2) CLP, (3) CLP + OS (20 mg/kg), and (4) CLP + MF (200 mg/kg MF ethanol extract). To evaluate the effects of MF and OS on the survival of CLP mice, MF and OS were administered orally for 3 days before the CLP procedure. Mouse survival was observed every 8 h for 4 days.

### 4.4. Cecal Ligation and Puncture-Induced Sepsis Model

C57BL/6 mice (male, 8 weeks old, 20–24 g) were anesthetized with isoflurane (5% induction, 2% maintenance); their lower abdomens were shaved and disinfected with alcohol prep pads. A surgical procedure involving a midline incision was conducted to locate and exteriorize the cecum. Using a 6-0 nylon (W1610T; Ethicon, Inc., Cincinnati, OH, USA), the cecum was tied off at 30% of its overall length. A 21-gauge needle was used to puncture the cecum, and a small quantity of feces was expelled to verify patency. Finally, the cecum was placed back in the abdominal cavity, and the surgical incisions were closed with metallic Reflex 7 mm clips (RS-9250; Michel Roboz Surgical, Inc., Gaithersburg, MD, USA). The mice were fluid resuscitated through subcutaneously administering pre-warmed normal saline (36 °C, 4 mL/100 g body weight). Sham group mice were subjected to cecal manipulation without ligation or puncture. All mice were administered an analgesic (tramadol, i.p., 10 mg/kg) twice daily. For survival analysis, the mortality rate in each group was observed for 4 days.

### 4.5. Assessment of Platelet Desialylation

Platelet isolation and washing were performed as previously described [54,55,56]. Briefly, 1 mL of peripheral blood was collected from the inferior vena cava and added to tubes containing 100 μL of ACD. The collected blood was then diluted with modified Tyrode’s HEPES buffer (pH 7.4) supplemented with 0.25 μM PGE1. The diluted sample was centrifuged at 150× *g* for 10 min and the resulting supernatant was collected and subjected to further centrifugation at 1200× *g* for 10 min. The platelet pellet was washed two times with Tyrode buffer, and then 10^6^ platelets were resuspended in 100 μL of PBS. The washed platelets were exposed to 5 mU neuraminidase from *C. perfringens* for 30 min at 37 °C and washed twice with modified Tyrode’s HEPES buffer. The cells were labeled with fluorescein isothiocyanate-labeled RCA-I, ECA, MAL-II, and SNA for 30 min at 24 °C. The labeled cells were then washed twice and analyzed using the Attune NxT flow cytometer (Thermo Fisher Scientific). FlowJo version 10 (Becton, Dickinson & Company, Franklin Lakes, NJ, USA) was used to analyze the data. 

### 4.6. Platelet Activation Assay

Platelet activation was assessed via measuring CD62P (P-selectin)-positive cells using flow cytometry as previously described [21,57]. Briefly, the washed platelets were incubated with 2.5 mU neuraminidase for 30 min at 37 °C, and then stimulated with 125 μM ADP. CD62P (p-selectin, platelet activation marker) and CD41 (platelet identification marker) antibodies were used to label the cells. Platelets were identified and gated via forward- and side-scatter distribution and CD41 positivity. Next, CD62P positivity for CD41-positive events was assessed.

### 4.7. Sialidase Activity Assay

Neuraminidase activity in mouse serum was measured using the Amplex Red Neuraminidase Assay Kit (Thermo Fisher Scientific), following the manufacturer’s instructions.

### 4.8. Hematological Analysis

Blood was collected 24 h after CLP and analyzed within 4 h of collection. Hemavet 950 (Drew Scientific, Waterbury, CT, USA) was used to measure hematologic parameters (Appendix A).

### 4.9. Measurement of Cytokines

Serum was separated from the blood samples for cytokine analysis and stored at −45 °C until measurement. The levels of serum IL-6 and TNF-α were measured using commercial ELISA kits following the manufacturer’s instructions.

### 4.10. Immunoblotting

Mouse livers were collected 24 h after CLP, flash-frozen in liquid nitrogen, and stored in a −45 °C freezer until analysis. The liver tissues were homogenized at a 1:10 ratio with T-PER™ Tissue Protein Extraction Reagent (Thermo Fisher Scientific) with protease and phosphatase inhibitor cocktails. The protein concentration of each sample was measured using Quick Start™ Bradford 1× Dye Reagent (Bio-Rad Laboratories, Richmond, CA, USA). Thirty micrograms of each protein sample was separated using 4–12% sodium dodecyl sulfate-polyacrylamide gel electrophoresis and then transferred onto polyvinyl difluoride (PVDF) membranes. The PVDF membranes were blocked with SuperBlock™ Blocking Buffer (Thermo Fisher Scientific) and incubated with primary antibodies at 4 °C overnight. The target proteins were identified using immunoblotting with specific antibodies against JAK2, phospho-JAK2, STAT3, and phospho-STAT3. The membranes were washed thrice using tris-buffered saline (TBST) with 0.05% tween 20, and then incubated with HRP-conjugated secondary antibody at a dilution of 1:10,000 for 1 h. The membranes were washed thrice with TBST. Protein bands were detected using the ECL kit (Thermo Fisher Scientific), and signal intensities were detected and visualized using the C-DiGit blot scanner (LI-COR Biosciences, Lincoln, NE, USA).

### 4.11. Hematoxylin and Eosin Staining and Histological Examination

The mouse lung and liver tissues were fixed in 10% neutral-buffered formalin. The fixed tissues were dehydrated in various concentrations of ethanol (70–100%) and then cleared using xylene. The tissues were then embedded in paraffin, sectioned to 4-μm thickness, and attached to slides. The tissue slides were stained with H&E, and quantitative analysis of tissue damage and inflammation was performed in a blinded manner using a light microscope (Leica Microsystems, Wetzlar, HE, Germany). Lung morphological changes were scored as no lesions (0), mild (1), moderate (2), or severe (3), based on the presence of hyperemia or congestion, neutrophil infiltration, alveolar hemorrhage, and cellular hyperplasia [36]. Liver morphological changes were scored as no lesions (0), mild (1), moderate (2), or severe (3), based on the presence of vacuolization, swollen hepatocytes, inflammation, and hepatocellular necrosis [37]. The sums of scores are presented as mean ± standard deviation (SD).

### 4.12. Immunohistochemistry

Immunohistochemical staining was carried out according to the manufacturer’s instructions, using the POLYVIEW^®^ PLUS HRP-DAB (Anti-Rabbit) Kit. Briefly, the lung and liver tissue sections were deparaffinized via treatment with xylene and rehydrated with decreasing concentrations of ethanol (50–100%) and then with deionized water. The sections were immersed in 1× antigen retrieval solution and subjected to 30 min heat-induced epitope retrieval in a dry oven preheated to 99 °C. Fibrin deposition was evaluated via incubating the sections with primary rabbit anti-fibrinogen/fibrin polyclonal antibody (1:200 dilution) overnight at 4 °C, and then with horseradish peroxidase (HRP)-DAB treatment and finally counterstaining with hematoxylin. The area of positive expression was identified in a blinded fashion using ImageJ software (NIH, Bethesda, MD, USA).

### 4.13. Gene Expression Analysis Using RT-qPCR

After collecting blood samples, the mouse livers were isolated and washed with PBS, and then stored at −45 °C until total RNA extraction. Twenty micrograms of the liver tissue was extracted using the RNeasy Mini Kit (Qiagen), and cDNA was synthesized using 1 μg of total RNA as a template using the iScript™ cDNA Synthesis Kit (Bio-rad). To quantify cDNA samples, real-time PCR was performed using SsoAdvanced Universal SYBR Green Supermix (Bio-Rad) and CFX96™ Real-Time System. The value of cycle threshold of TPO and cyclophilin A (cycloA) was measured using Bio-Rad CFX Manager. The expression of target genes relative to that of cycloA was determined using the 2−ΔΔCT method. All primers used in this study are listed in Appendix A.

### 4.14. Data and Statistical Analysis

The data are expressed as mean ± SD. The results were statistically analyzed using a one-way ANOVA, followed by Tukey’s multiple comparison test. The survival rate of mice was analyzed using the log-rank (Mantel–Cox) test. GraphPad Prism 7 software (San Diego, CA, USA) was used for all statistical analyses. Results with *p* < 0.05 were considered to be statistically significant.

## 5. Conclusions

The present study confirmed that MF inhibits sialidase activity and prevents platelet-desialylation-induced SAT and platelet activation. The extract effectively reduced platelet activation and desialylation in vitro and in vivo. Moreover, MF increased survival, reduced the inflammatory response, and maintained circulating platelet levels in the CLP-induced sepsis model. In addition, MF downregulated hepatic AMR-mediated platelet clearance and JAK2 and STAT3 phosphorylation through reducing platelet desialylation. Collectively, these findings suggest that MF could be a promising therapeutic option for sepsis and SAT.

## Figures and Tables

**Figure 1 ijms-24-08863-f001:**
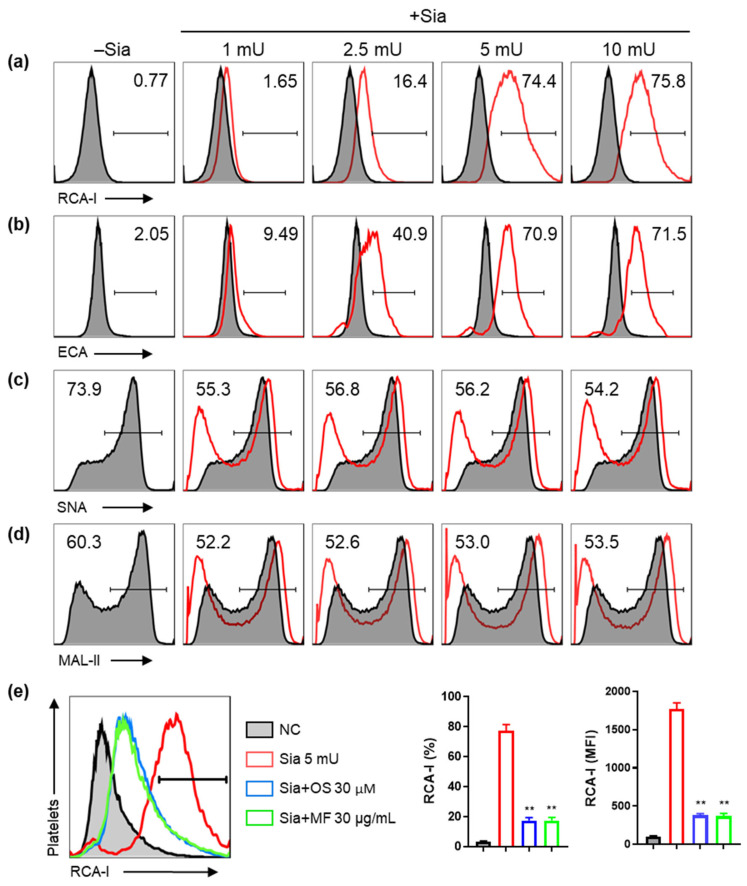
RCA-I, ECA, SNA, and MAL-II binding and inhibitory effects of MF on platelet desialylation in washed platelets exposed to sialidase. Washed platelets were subjected to treatment without sialidase (−Sia) or with varying concentrations of sialidase (+Sia; 1, 2.5, 5, and 10 mU) for 30 min at 37 °C. Following the treatment, the binding of platelets to (**a**) RCA-I, (**b**) ECA, (**c**) SNA, and (**d**) MAL-II was confirmed using flow cytometry (FACS). (**e**) Washed platelets were treated with sialidase and MF or OS, and RCA-I binding was measured. Representative FACS data from one of three separate experiments are shown. The bar graphs presented indicate mean ± standard deviation (SD) of three samples. All experiments were repeated thrice, and similar outcomes were observed. ** *p* < 0.01 compared with the Sia 5 mU group. RCA-I, *Ricinus communis*-1 agglutinin lectin; ECA, *Erythrina crista-galli* lectin; SNA, *Sambucus nigra* lectin; MAL-II, *Maackia amurensis* lectin II; MFI, mean fluorescence intensity; MF, *Myristica fragrans*; OS, oseltamivir phosphate; NC, negative control.

**Figure 2 ijms-24-08863-f002:**
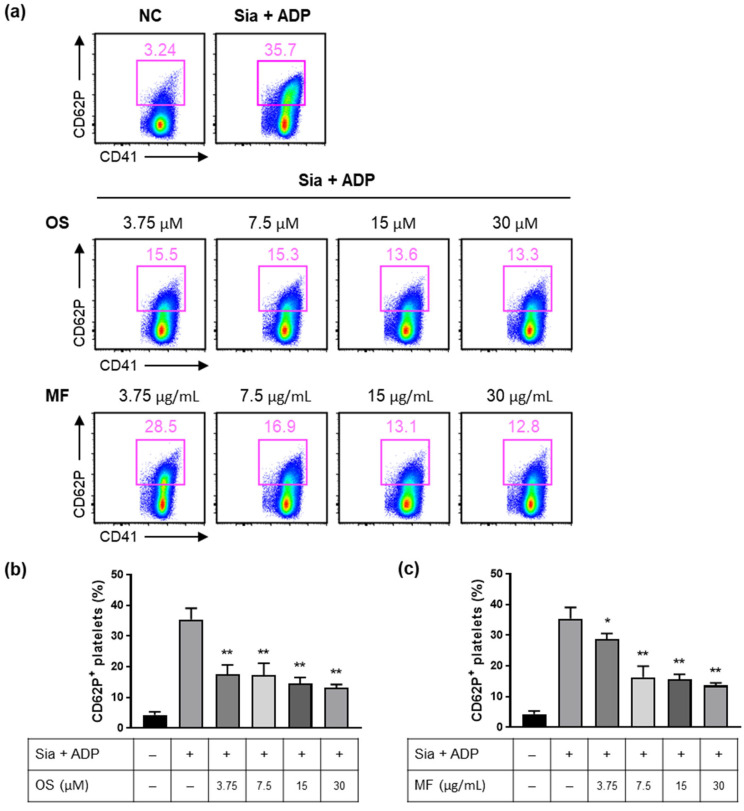
Effects of MF on the suppression of sialidase-mediated platelet hyper-reactivity and activation in washed platelets. (**a**) CD62P expression in washed platelets was measured after stimulation with 5 mU sialidase and 125 μM ADP in the presence of MF or OS at different concentrations. Representative FACS data from one of three separate experiments are shown. (**b**) Inhibition of CD62P expression using OS in Sia + ADP-stimulated washed platelets. (**c**) Inhibition of CD62P expression using MF in Sia + ADP-stimulated washed platelets. The bar graphs presented indicate mean ± standard deviation (SD) of three samples. All experiments were repeated thrice, and similar outcomes were observed. * *p* < 0.05 and ** *p* < 0.01 compared with the Sia + ADP group. ADP, adenosine diphosphate; MF, *Myristica fragrans*; OS, oseltamivir phosphate; Sia, sialidase.

**Figure 3 ijms-24-08863-f003:**
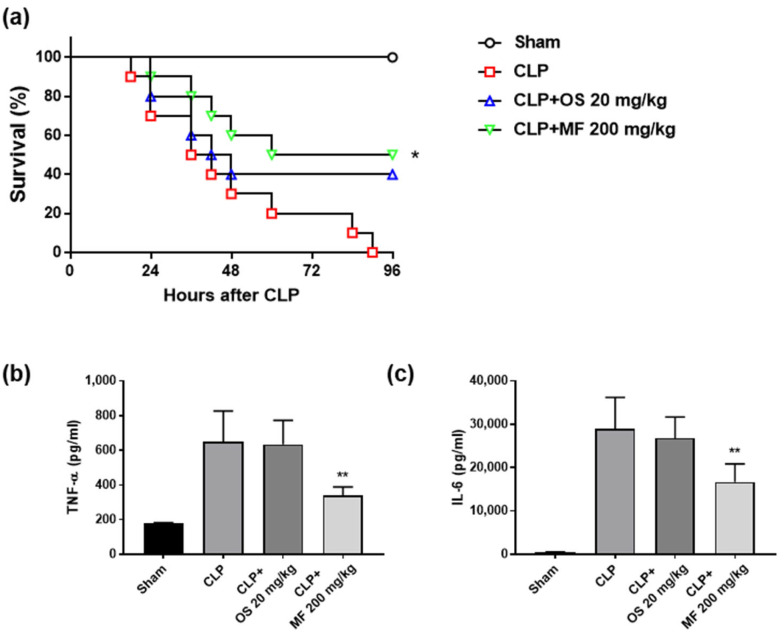
Effects of MF on the survival of and inflammation in cecal ligation and puncture (CLP)-induced septic mice. (**a**) MF (200 mg/kg) and OS (20 mg/kg) were administered orally for 3 days before the CLP procedure, and survival was monitored every 8 h daily for up to 4 days after CLP. (**b**,**c**) Serum TNF-α and IL-6 levels in each group were measured using ELISA. Values are presented as mean ± SD (n = 5 mice). * *p* < 0.05 and ** *p* < 0.01 compared with the CLP group. MF, *Myristica fragrans*; OS, oseltamivir phosphate.

**Figure 4 ijms-24-08863-f004:**
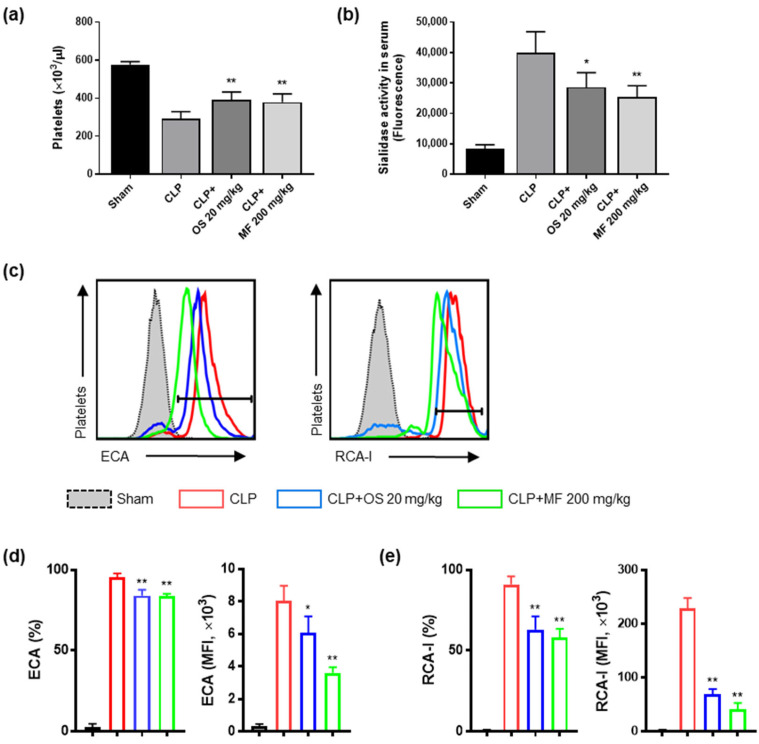
Effect of MF on platelet desialylation and thrombocytopenia in cecal ligation and puncture (CLP)-induced septic mice. (**a**) Circulating platelet count. (**b**) Sialidase activity was measured in the serum of each group of mice. (**c**) Platelets isolated and washed from mice in each group were stained for CD41, ECA, and RCA-I, and platelet desialylation was analyzed using flow cytometry. (**d**) Bar graphs for the percentage of ECA binding and MFI on platelet surface in each group. (**e**) Bar graphs for the percentage of RCA-I binding and MFI on platelet surface in each group. The data are presented as mean ± standard deviation (SD) with a sample size of five mice per group. Representative FACS data from one of three separate experiments are shown. * *p* < 0.05 and ** *p* < 0.01 compared with the CLP group. MF, *Myristica fragrans*; OS, oseltamivir phosphate; ECA, *Erythrina crista-galli* lectin; RCA-I, *Ricinus communis*-1 agglutinin lectin.

**Figure 5 ijms-24-08863-f005:**
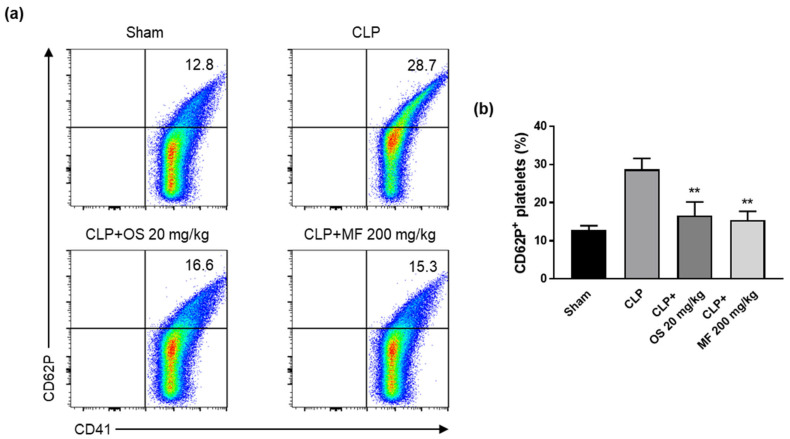
Effect of MF on the suppression of platelet activation in mice with CLP-induced sepsis. (**a**) Platelet activation was determined based on CD41 and CD62P (P-selectin) expression. (**b**) Bar graphs for the percentage of CD62P-positive platelets in each group. Percentages of CD62P-positive platelets are presented as mean ± SD (n = 5 mice). Representative FACS data from one of three separate experiments are shown. ** *p* < 0.01 compared with the CLP group. MF, *Myristica fragrans*; OS, oseltamivir phosphate.

**Figure 6 ijms-24-08863-f006:**
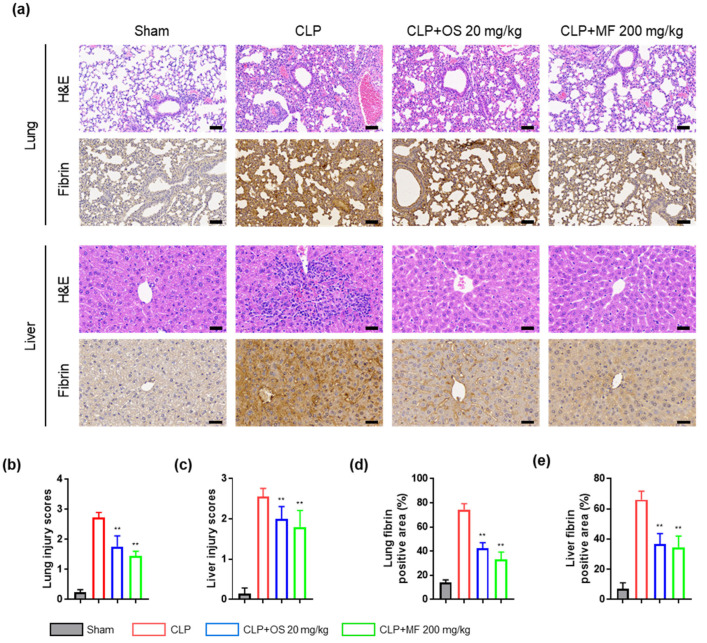
Effects of MF on organ damage and fibrin deposition in CLP-induced septic mice. (**a**) Histopathological analysis of lung and liver tissues was performed using H&E and immunohistochemical staining to determine inflammation and fibrin deposition. Representative sections are shown at ×200 (lung section) and ×400 (liver section) magnifications; scale bar: 60 and 30 μm, respectively. Representative images from each group were selected (n = 5). (**b**) Bar graphs for the lung injury scores of each group. Lung damage was assessed via scoring necrosis, inflammatory cell infiltration, hemorrhage, and congestion. (**c**) Bar graphs for liver injury scores of each group. Liver damage was assessed via scoring vacuolization, swollen hepatocytes, hepatocellular necrosis, and inflammatory cell infiltration. (**d**) Bar graphs for lung fibrin deposition scores. (**e**) Bar graphs for liver fibrin deposition scores. Semi-quantitative analysis of immunohistochemical images using ImageJ software was performed. The data are presented as mean ± standard deviation (SD) with a sample size of five mice. ** *p* < 0.01 compared with the CLP group. MF, *Myristica fragrans*; OS, oseltamivir phosphate.

**Figure 7 ijms-24-08863-f007:**
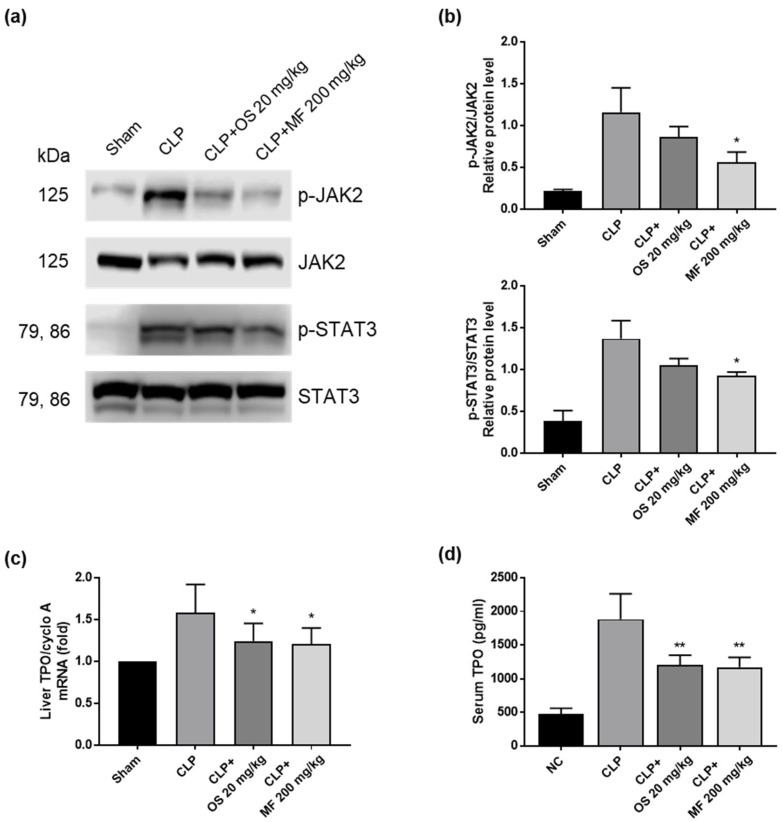
Effects of MF on hepatic JAK2/STAT3 signaling and TPO expression in CLP-induced septic mice. (**a**) Phosphorylated JAK2 and STAT3 levels in the liver tissue were analyzed using Western blotting (**b**) Western blot signals of phosphorylated JAK2 and STAT3 were normalized to those of total JAK2 and STAT3. Relative expression of phosphorylated JAK2 and STAT3 was quantified based on three independent experiments. (**c**) The expression of liver TPO mRNA was determined using qPCR. (**d**) The serum TPO level was quantified using ELISA. The data are presented as mean ± standard deviation (SD) with a sample size of five mice. * *p* < 0.05 and ** *p* < 0.01 compared with the CLP group. MF, *Myristica fragrans*; OS, oseltamivir phosphate; JAK2, Janus kinase 2; STAT3, signal transducer and activator of transcription 3; TPO, thrombopoietin.

**Table 1 ijms-24-08863-t001:** Major composition of EtOH extract of *M. fragrans*.

No.	Compound	Molecular Formula	Molecular Weight (g/moL)
1	3,5-Dihydroxyestragole	C_16_H_22_O_8_	342.3
2	Methoxyeugenol	C_11_H_14_O_3_	194.2
3	Myristicin	C_11_H_12_O_3_	192.2
4	Myrislignan	C_21_H_26_O_6_	374.4
5	Myrislignanometin E	C_21_H_26_O_7_	390.4
6	Maceneolignan H	C_24_H_30_O_7_	430.5
7	Licarin A	C_20_H_22_O_4_	326.4
8	Licarin B	C_20_H_20_O_4_	324.4
9	5′-Methoxylicarin B	C_21_H_22_O_5_	354.3
10	Verrucosin	C_20_H_24_O_5_	344.4
11	Malabaricone B	C_21_H_26_O_4_	342.4
12	Malabaricone C	C_21_H_26_O_5_	358.4

## Data Availability

The datasets used and/or analyzed during the current study are available from the corresponding author on reasonable request.

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
