# Peer review of "Myristica fragrans Extract Inhibits Platelet Desialylation and Activation to Ameliorate Sepsis-Associated Thrombocytopenia in a Murine CLP-Induced Sepsis Model"

_ijms, 2023, doi:10.3390/ijms24108863_

Round 1

Reviewer 1 Report

The authors demonstrate in this study, that the sialidase inhibitory activity of Myristica fragrans (MF) extracts is effective for treating sepsis and Sepsis-associated thrombocytopenia (SAT). Sialidases are important enzymes that remove sialic acid. The authors earlier reported the efficacy of MF as a natural sialidase inhibitor [ref 30 in the paper) and now studied the effects of MF on sepsis and SAT, via inhibiting sialidase-mediated platelet desialylation.

The authors convincingly show in the well documented study that the plant-derived MF extract has efficacy in treating sepsis and SAT in the murine CLP-induced sepsis model. The data are of wide interest and clearly deserve publication. This referee has only a few questions/ comments which the authors may want to address.

a)  What is the advantage of MF over oseltamivir (OS), which they also used?

b)  Have they compared the MF effects with effects of platelet inhibitors (P2Y12, Syk, Btk others)

c)  What would be the next steps(s) to show that their present information is important also for humans ?

ok

Author Response

Responses to the Comments of Reviewer 1

Comments and Suggestions to the Authors

The authors demonstrate in this study, that the sialidase inhibitory activity of Myristica fragrans (MF) extracts is effective for treating sepsis and Sepsis-associated thrombocytopenia (SAT). Sialidases are important enzymes that remove sialic acid. The authors earlier reported the efficacy of MF as a natural sialidase inhibitor [ref 30 in the paper) and now studied the effects of MF on sepsis and SAT, via inhibiting sialidase-mediated platelet desialylation.

The authors convincingly show in the well documented study that the plant-derived MF extract has efficacy in treating sepsis and SAT in the murine CLP-induced sepsis model. The data are of wide interest and clearly deserve publication. This referee has only a few questions/ comments which the authors may want to address.

a) What is the advantage of MF over oseltamivir (OS), which they also used?

Response: Both OS and MF are sialidase inhibitors. However, OS is an antiviral agent with viral sialidase inhibitory activity and MF is an extract with bacterial sialidase inhibitory activity. According to our previous findings, MF exhibits inhibitory activity against bacterial sialidases Nan A, B, and C [1]. In addition, the antibacterial, anti-inflammatory, and antioxidant effects of MF have been reported previously [2,3]. In this study, the bacterial sialidase inhibitory effect of MF, an extract, was equivalent to that of OS, suggesting the potential of MF as a natural medicine for sepsis.

[1] Park, J.-Y.; Hwan Lim, S.; Ram Kim, B.; Jae Jeong, H.; Kwon, H.-J.; Song, G.-Y.; Bae Ryu, Y.; Song Lee, W. Sialidase inhibitory activity of diarylnonanoid and neolignan compounds extracted from the seeds of Myristica fragrans. Bioorganic & Medicinal Chemistry Letters 2017, 27, 3060-3064.

[2] Jaiswal, P.; Kumar, P.; Singh, V.K.; Singh, D.K. Biological effects of Myristica fragrans. Annual Review of Biomedical Sciences 2009, 11.

[3] Ozaki, Y.; Soedigdo, S.; Wattimena, Y.R.; Suganda, A.G. Antiinflammatory effect of mace, aril of Myristica fragrans Houtt., and its active principles. Japanese Journal of Pharmacology 1989, 49, 155-163.

b) Have they compared the MF effects with effects of platelet inhibitors (P2Y12, Syk, Btk others)

Response: P2Y12, Syk, and Btk are well-known inhibitors of platelet activation. However, in this study, we aimed to confirm the platelet-inhibitory activity according to desialylation, and OS, a representative inhibitor of sialidase activity, was used as a positive control. We will consider the suggested inhibitors when conducting research related to platelet inhibition.

c) What would be the next steps(s) to show that their present information is important also for humans?

Response: Based on the results of this study, to determine the applicability of MF to treat human sepsis and SAT, we plan to confirm the prevention of desialylation of human platelets with MF and the inhibition of platelet phagocytosis by human hepatocyte cells (hepG2).

Reviewer 2 Report

In this manuscript Jeong et al investigated ethanol extract Myristica fragrans (MF) on mouse sepsis model and protective effects of MF on sepsis-associated thrombocytopenia (SAT). They showed that MF inhibited platelet desialylation and activation by inhibiting bacterial sialidase and improved survival and and inflammation in a mouse model cecal ligation and puncture (CLP)-induced sepsis. MF also reduces hepatic platelet clearance, hepatic JAK2/STAT3 phosphorylation and thrombopoietin mRNA expression. All experiments are clear and very well performed. The conclusions are correct and I have no questions to the experimental part of the manuscript. However, the scientific significance of the presented results are under question because:

1.      That sepsis and other bacterial infections increases platelet desialylation which lead to enhanced platelet clearance and thrombocytopenia is well-known.

2.      The sialidase inhibitor, oseltamivir phosphate inhibited platelet desialylation and prevented thrombocytopenia is known.

3.      In the previous manuscript from the same group it was shown that compounds from MF are potential sialidase inhibitors. Moreover, they showed that malabaricone C is the most potent pneumococcal sialidases inhibitor. It is not clear why, instead of ethanol extract, this compound was not used in this study.

4.      Most of the presented results are simply predicted because it is well-known the consequences of platelet desialylation inhibition that it will increase platelet count reduce platelet activation organ damage and TPO concentration which is connected with the activity of JAK/STAT signaling pathway.

5.      The only differences between MF and oseltamivir phosphate was connected with inhibition of inflammatory cytokine (TNF-α and IL-6) levels by MF but not by oseltamivir. The authors, even in the discussion, did not explained this interesting and important observation.

6.      The authors claim that MF-induced platelet inhibition is connected with inhibition of platelet desialylation, however previously (Phytotherapy Res 2013) platelet inhibition by one compound isolated from MF, independent from desialylation and, most probably, connected with activation of PKA was shown. Therefore, the inhibition of platelet activation probably is not connected only with prevention of desialylation. Such data should be included in the manuscript.      

Author Response

Responses to the Comments of Reviewer 2

Comments and Suggestions for Authors

In this manuscript Jeong et al investigated ethanol extract Myristica fragrans (MF) on mouse sepsis model and protective effects of MF on sepsis-associated thrombocytopenia (SAT). They showed that MF inhibited platelet desialylation and activation by inhibiting bacterial sialidase activity and improved survival and and inflammation in a mouse model cecal ligation and puncture (CLP)-induced sepsis. MF also reduces hepatic platelet clearance, hepatic JAK2/STAT3 phosphorylation and thrombopoietin mRNA expression. All experiments are clear and very well performed. The conclusions are correct and I have no questions to the experimental part of the manuscript. However, the scientific significance of the presented results are under question because:

  1. That sepsis and other bacterial infections increases platelet desialylation which lead to enhanced platelet clearance and thrombocytopenia is well-known.

  1. The sialidase inhibitor, oseltamivir phosphate inhibited platelet desialylation and prevented thrombocytopenia is known.

Response to comments 1 and 2: We thank you for the comments. As you mentioned, it has been reported that sialidase inhibitors inhibit platelet desialylation and prevent thrombocytopenia [1]; therefore, we conducted this study using MF, a bacterial sialidase inhibitor.

[1] Li, M.F.; Li, X.L.; Fan, K.L.; Yu, Y.Y.; Gong, J.; Geng, S.Y.; Liang, Y.F.; Huang, L.; Qiu, J.H.; Tian, X.H., et al. Platelet desialylation is a novel mechanism and a therapeutic target in thrombocytopenia during sepsis: An open-label, multicenter, randomized con-trolled trial. Journal of Hematology & Oncology 2017, 10, 104.

  1. In the previous manuscript from the same group it was shown that compounds from MF are potential sialidase inhibitors. Moreover, they showed that malabaricone C is the most potent pneumococcal sialidases inhibitor. It is not clear why, instead of ethanol extract, this compound was not used in this study.

Response: This study was performed with MF instead of malabaricone C (Mal C) for the following reasons:

1. In our previous study, Mal C exhibited high inhibitory activity against Nan A, B, and C secreted by Streptococcus pneumoniae.

2. This study confirmed the platelet activation-inhibitory effect of MF based on its bacterial sialidase inhibitory activity.

3. As for Mal C, we are conducting a study on sepsis caused by pneumococci.

  1. Most of the presented results are simply predicted because it is well-known the consequences of platelet desialylation inhibition that it will increase platelet count reduce platelet activation organ damage and TPO concentration which is connected with the activity of JAK/STAT signaling pathway.

Response: As you mentioned, the mechanisms of thrombocytopenia mediated by platelet desialylation are well known. We conducted this study to investigate the effect of MF on such mechanisms.

  1. The only differences between MF and oseltamivir phosphate was connected with inhibition of inflammatory cytokine (TNF-α and IL-6) levels by MF but not by oseltamivir. The authors, even in the discussion, did not explained this interesting and important observation.

Response: We agree with your comment. Per your suggestion, we have discussed the suppression of inflammatory cytokine (TNF-α and IL-6) expression by MF in the revised manuscript (Discussion: page 11, lines 296–297).

“These results indicate the anti-inflammatory activity of MF, which is consistent with the findings of a previous study [23].”

  1. The authors claim that MF-induced platelet inhibition is connected with inhibition of platelet desialylation, however previously (Phytotherapy Res 2013) platelet inhibition by one compound isolated from MF, independent from desialylation and, most probably, connected with activation of PKA was shown. Therefore, the inhibition of platelet activation probably is not connected only with prevention of desialylation. Such data should be included in the manuscript.

Response: We thank you for your thoughtful comment. Per your suggestion, the previously reported platelet activation-inhibitory effect of neolignans isolated from MF has been discussed in the manuscript (Discussion: page 11, lines 289–292).

“Additionally, in a previous study, a neolignan isolated from MF exhibited antiplatelet activity by regulating the cAMP level [49]. Therefore, these results demonstrate the potential of MF as an inhibitor of platelet activation.”

Reviewer 3 Report

This is a well-written, well-designed and well-executed study that is of significance to sepsis, which is a critical cause of morbidity and mortality worldwide. We have a few comments that I think will improve the manuscript if addressed by the authors:

1. The authors should comment about the safety of MF, and potentially the ingredient in MF that is responsible for the pharmacological activity. The authors describe the various components but there is no discussion of which one(s) could be producing the effect. Do they have any data they could share?

2. The authors should also assess the effect of MF on Netosis, as that's a critical part of the characterization. 

3. Please provide a comparison between MF and OS and how the application/use of each will be different.

There are some minor typos that need to be fixed

Author Response

Responses to the Comments of Reviewer 3

Comments and Suggestions for Authors

This is a well-written, well-designed and well-executed study that is of significance to sepsis, which is a critical cause of morbidity and mortality worldwide. We have a few comments that I think will improve the manuscript if addressed by the authors:

  1. The authors should comment about the safety of MF, and potentially the ingredient in MF that is responsible for the pharmacological activity. The authors describe the various components but there is no discussion of which one(s) could be producing the effect. Do they have any data they could share?

Response: We thank you for the comment. MF has traditionally been used as a spice and medicine. As previously reported, MF exerts pharmacological effects including anti-inflammatory, antibacterial, antioxidant, and analgesic effects [1,2]. In our previous study, we identified 12 compounds by isolating and purifying MF, and reported that these compounds have sialidase inhibitory activity [3].

[1] Jaiswal, P.; Kumar, P.; Singh, V.K.; Singh, D.K. Biological effects of Myristica fragrans. Annual Review of Biomedical Sciences 2009, 11.

[2] Ozaki, Y.; Soedigdo, S.; Wattimena, Y.R.; Suganda, A.G. Antiinflammatory effect of mace, aril of Myristica fragrans Houtt., and its active principles. The Japanese Journal of Pharmacology 1989, 49, 155-163.

[3] Park, J.-Y.; Hwan Lim, S.; Ram Kim, B.; Jae Jeong, H.; Kwon, H.-J.; Song, G.-Y.; Bae Ryu, Y.; Song Lee, W. Sialidase inhibitory activity of diarylnonanoid and neolignan compounds extracted from the seeds of Myristica fragrans. Bioorganic & Medicinal Chemistry Letters 2017, 27, 3060-3064.

  1. The authors should also assess the effect of MF on Netosis, as that's a critical part of the characterization.

Response: We agree with the comment. However, the purpose of our study was to determine whether MF prevents platelet desialylation by inhibiting sialidase activity. Platelet desialylation increases the sensitivity to platelet activation by platelet agonists such as ADP. Therefore, only platelet activation through p-selectin expression measurement was investigated. We will consider your suggestion in our future research.

  1. Please provide a comparison between MF and OS and how the application/use of each will be different.

Response: Both OS and MF are sialidase inhibitors. However, OS is an antiviral drug used to treat and prevent influenza virus A and B infection. In this study, MF was confirmed to have bacterial sialidase inhibitory activity, and also antibacterial, anti-inflammatory, and antioxidant effects; therefore, it has an advantage in suppressing bacterial sepsis.

Round 2

Reviewer 3 Report

I am satisfied with the author's response.